# A puzzling insensitivity of magnon spin diffusion to the presence of 180-degree domain walls

Ruofan Li[1], Lauren J. Riddiford[2], Yahong Chai[3], Minyi Dai[4], Hai Zhong [5], Bo Li[6], Peng Li[2], Di Yi [2,7], Yuejie Zhang [3], David A. Broadway[8], Adrien E. E. Dubois[5,8], Patrick Maletinsky[5,8], Jiamian Hu [4], Yuri Suzuki [2,9], Daniel C. Ralph [1,10] ✉ & Tianxiang Nan [1,3] ✉

We present room-temperature measurements of magnon spin diffusion in epitaxial ferrimagnetic insulator $MgAl_{0.5}Fe_{1.5}O_4$ (MAFO) thin films near zero applied magnetic field where the sample forms a multi-domain state. Due to a weak uniaxial magnetic anisotropy, the domains are separated primarily by 180° domain walls. We find, surprisingly, that the presence of the domain walls has very little effect on the spin diffusion – nonlocal spin transport signals in the multi-domain state retain at least 95% of the maximum signal strength measured for the spatially-uniform magnetic state, over distances at least five times the typical domain size. This result is in conflict with simple models of interactions between magnons and static domain walls, which predict that the spin polarization carried by the magnons reverses upon passage through a 180° domain wall.

Magnons offer the intriguing potential to transport and process information with low dissipation and in samples with non-volatility[1,2]. Diffusive flows of incoherent magnons can be measured in a nonlocal geometry where spin injection into a magnetic insulator is driven by the spin Hall effect and spin Seebeck effect at one heavy metal/magnetic insulator interface, and the resulting spin currents can then be detected by the inverse spin Hall effect at a second distant heavy metal/magnetic insulator interface[3,4]. Propagation of magnon spin currents over long distances in such structures is now routinely realized in ferrimagnetic[3–14] and antiferromagnetic insulators[15–18].

If magnon spin transport experiments are performed with sufficiently small magnetic fields that the magnetic material forms domains, it is important to consider what happens to the flow of angular momentum at magnetic domain walls[19–24]. Assuming non-interacting magnons and in the limit that a magnon has a short wavelength compared to the width of a domain wall, a magnon is predicted to propagate through the domain wall without scattering, while the angular momentum carried by the magnon rotates adiabatically within the texture of the domain wall[25,26]. The angular momentum carried by a current of short-wavelength magnons should therefore be reversed upon transiting a 180° domain wall. In a sample with many domains, in which some paths between electrodes transit an even number of domain walls and some an odd number, in this picture the total spin current carried by magnons should decay to zero on a length scale comparable to the domain size. For longer-wavelength magnons, with wavelengths comparable to or greater than the domain wall width, magnons should also partially backscatter in addition to reversing their spin direction upon transmission through

[1]Laboratory of Atomic and Solid-State Physics, Cornell University, Ithaca, NY 14853, USA. [2]Geballe Laboratory for Advanced Materials, Stanford University, Stanford, CA 94305, USA. [3]School of Integrated Circuits and Beijing National Research Center for Information Science and Technology (BNRist), Tsinghua University, 100084 Beijing, China. [4]Department of Materials Science and Engineering, University of Wisconsin-Madison, Madison, WI 53706, USA. [5]Qnami AG, CH-4132 Muttenz, Switzerland. [6]Institute for Advanced Study, Tsinghua University, 100084 Beijing, China. [7]State Key Lab of New Ceramics and Fine Processing, School of Materials Science and Engineering, Tsinghua University, 100084 Beijing, China. [8]Department of Physics, University of Basel, CH-4056 Basel, Switzerland. [9]Department of Applied Physics, Stanford University, Stanford, CA 94305, USA. [10]Kavli Institute at Cornell for Nanoscale Science, Ithaca, NY 14853, USA. ✉e-mail: dcr14@cornell.edu; nantianxiang@mail.tsinghua.edu.cn

the domain wall[27–31]. In both limits, therefore, the presence of 180° domain walls is expected to suppress the net flow of diffusive spin currents compared to a uniform magnetic state without domain walls. Nonlocal spin transport experiments performed in antiferromagnets have confirmed that the effective spin diffusion length is much shorter in samples with small domains having effectively randomly-oriented Néel vectors compared to samples with large domain sizes[18].

Here we measure the diffusive flow of incoherent magnons at room temperature in a ferrimagnetic insulator thin film which has sufficient uniaxial anisotropy that the magnetic domains near zero applied magnetic field are aligned primarily along this axis of uniaxial anisotropy, producing a distribution of primarily 180° domain walls. We find, quite surprisingly, that the presence of these domain walls produces very little effect on the diffusion of spin angular momentum. When the spin polarization associated with the spin Hall effect in a Pt injector wire is aligned with the anisotropy axis, we measure nonlocal spin transmission signals that are at least 95% as large as for a completely-saturated magnetic sample, even when a typical path between injector and detector electrodes must pass through at least 5 domain walls. This represents a striking conflict with the standard models of interactions between magnons and static domain walls. We present some speculative suggestions for how these models might be generalized to explain this result.

## Results

Specifically, we measure non-local magnon spin transport in a low-damping spinel ferrite, $MgAl_{0.5}Fe_{1.5}O_4$ (MAFO), which is in the form of an ultra-thin (6 nm) epitaxial film on a $MgAl_2O_4$ substrate[32–36]. We use samples in which Pt injector and detector wires are separated by distances ranging from 0.4 to 4.2 μm. Previously, we have reported magnon propagation over micrometer distances in such films and observed larger propagation lengths (by >30%) along the four-fold <110> family of axes (easy axes) compared to the <100> axes (hard axes)[37]. Those measurements were performed under an external magnetic field that was high enough (75 mT) to always saturate the magnetization of the film in plane. At very low magnetic fields these same samples form magnetic domains, and in this domain structure we observe a small additional anisotropy between the two easy axes [110] and [1$\bar{1}$0]. This can be seen clearly from differences in the magnetic hysteresis loops measured with field applied along these two directions (Fig. 1a). (This measurement was performed on the same 6 nm film which was used for the rest of the experimental results.) For the harder axis between these two, i.e. along [110], the hysteresis curve contains a plateau near zero magnetization in the applied-field range between ±0.2 mT, indicating a stable axis for the spins along [1$\bar{1}$0]. For

a field sweep along the [1$\bar{1}$0] axis, the magnetization reverses in one jump with no intermediate step, as expected for an easy axis.

We have measured the size scale of the magnetic domains near zero applied magnetic field using scanning nitrogen-vacancy (NV) center magnetometry[38] in the multi-domain state. The sample was first saturated with a 50 mT field along the [110] direction. Upon removing the applied field, the stray magnetic field map from the sample was measured, and the magnetization distribution along [1$\bar{1}$0] (Fig. 1b) was determined by a reverse propagation algorithm[39] (see the Supplementary Note 1 for the measured stray magnetic field map and the algorithm used to construct the magnetization map). We observe a disordered stripe-like pattern aligned preferentially along the [1$\bar{1}$0] axis with typical lengths considerably <1 μm in the [1$\bar{1}$0] direction. (An autocorrelation analysis of Fig. 1b gives a correlation length of 200 nm along [1$\bar{1}$0].) Given the easy-axis determination from Fig. 1a, we assumed in the reverse propagation algorithm that the local magnetization is aligned along the [1$\bar{1}$0] or [$\bar{1}$10] directions with 180° domain walls in between. As a test, we also tried modeling the field map assuming easy axes along [100], along [110], or along [010], but these assumptions were not able to reproduce the measurement with the same low error (see Supplementary Fig. 1). We speculate that the weak uniaxial anisotropy favoring magnetization along the [1$\bar{1}$0] axes might result from anisotropic pinning sites generated during the growth process of the MAFO films by asymmetry in the deposition plume at the position of the sample or by step structure in the substrate (see the discussion of micromagnetic simulations in the Supplementary Note 2).

To study the magnon transport, we employed the nonlocal geometry shown schematically in Fig. 2a, where pairs of parallel Pt wires are deposited on top of the MAFO thin films to act as magnon injector and detector[3,5,8,9,13,15,17]. A scanning electron microscope image of a device is shown in Fig. 2b. A charge current/passing through the injector wire produces magnons in the magnetic film due to the spin Hall effect (SHE) in Pt, and also due to the spin Seebeck effect arising from a thermal gradient due to Joule heating[40]. The magnons diffuse through the film to the detector where they produce a voltage signal via the inverse spin Hall effect (ISHE)[41]. By using a lock-in amplifier, we distinguish between the signals from electrically and thermally injected magnons by detecting the first ($V_{1\omega}$) and second ($V_{2\omega}$) harmonic responses, from which the nonlocal resistances can be obtained as $R_{1\omega} = V_{1\omega}/I$ and $R_{2\omega} = V_{2\omega}/I^2$. Figure 2c shows the nonlocal resistances with the Pt wires perpendicular to the [100] axis, as a function of the angle $\phi$ between the magnetization and Pt wires with a large enough field magnitude (75 mT) that the magnetization remains fully saturated in-plane. For $R_{1\omega}$, both the injection (via SHE) and the detection (via

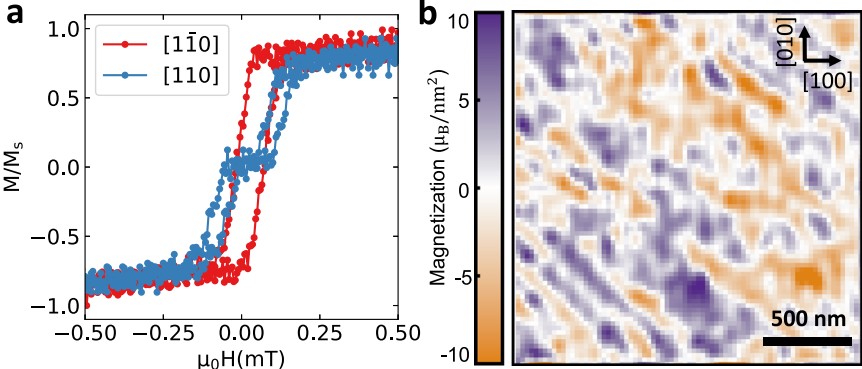

**Fig. 1 | Magnetic hysteresis and imaging of magnetization with the scanning NV magnetometry for MAFO thin films. a** Magnetic hysteresis loops measured with a SQUID magnetometer for a 6 nm MAFO film, which is the same film used for nonlocal spin transport measurements. The magnetic field is swept along the in-

plane [1$\bar{1}$0] and [110] axes. **b** Magnetization map obtained by reverse propagation of a scanning NV center stray magnetic field map of the MAFO thin film in the multidomain state at zero field after saturation in the [110] direction.

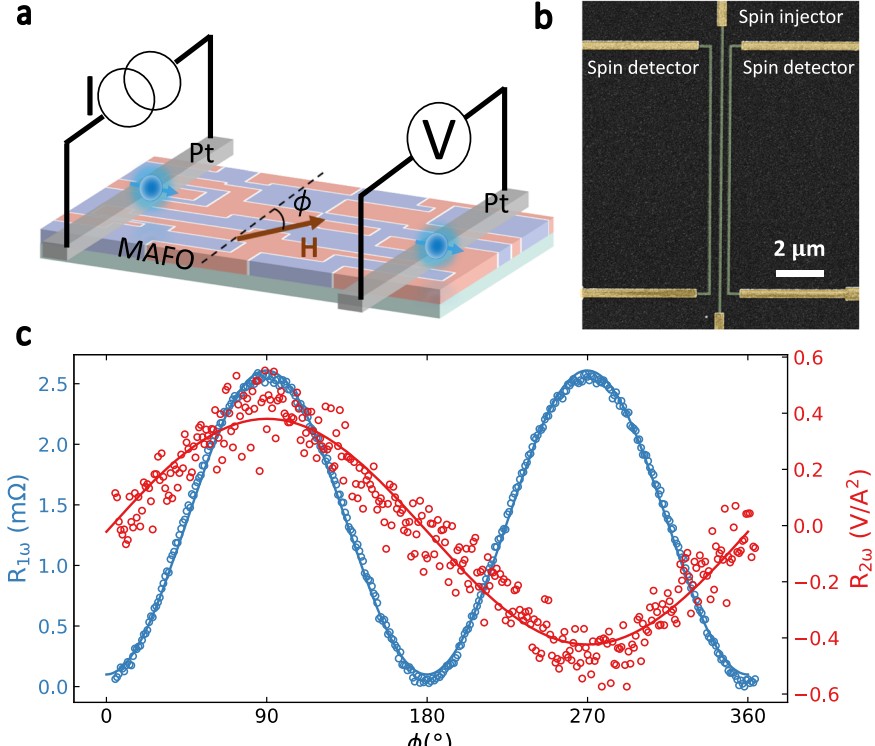

**Fig. 2 | Nonlocal magnon transport measurement. a** Schematic drawing of the experimental setup for nonlocal measurements. **b** False-color scanning electron microscope image of the device, in which green indicates the Pt wires and yellow corresponds to the contact leads. **c** First harmonic $R_{1\omega}$ and second-harmonic $R_{2\omega}$ nonlocal signals as functions of the magnetic-field angle $\phi$ for a field magnitude of 75 mT and for Pt wires oriented along the [010] axis with a 1 μm separation.

ISHE) of the magnons have a $\sin(\phi)$ dependence (because the magnon modes can be most effectively excited or annihilated when the spin polarization associated with the spin Hall effect is collinear with the magnetization), producing a total dependence of approximately $\sin^2(\phi)$. On the other hand, $R_{2\omega}$ shows an approximately $\sin(\phi)$ angular dependence, because the injection of the magnons (via SSE) has no angular dependence, while the detection of the magnons through ISHE varies as $\sin(\phi)$.

With the dependence of nonlocal resistances on uniform magnetization established, we systematically investigated magnon transport in the multi-domain state at low field, with magnons propagating along different crystal axes. Figure 3 shows the magnetic field dependence of normalized nonlocal resistances for the first and second harmonic signals, $R_{1\omega}$ and $R_{2\omega}$, with the Pt wires along [010], [1$\bar{1}$0], and [110] axes. During the measurements, the magnetic field $H_{ext}$ was swept either parallel ($\phi = 0°$) or perpendicular ($\phi = 90°$) to the Pt wires.

For all configurations, at fields strong enough to align the magnetic moment in the MAFO, $R_{1\omega}$ is at maximum (zero) when $\phi = 90°$ ($\phi = 0°$), following the $\sin^2(\phi)$ dependence, consistent with that shown in Fig. 2c. However, there are strong differences among the different device geometries in the low field regime, depending on the relative orientation between the magnon propagation and the crystal axes. As shown in Fig. 3a, when magnons propagate along the [100] axis, i.e. with Pt wires along the [010] (or [100]) axis, $R_{1\omega}$ decreases near zero field for $\phi = 90°$ while increasing for $\phi = 0°$, with a hysteresis behavior corresponding to the magnetization reversal. At the coercive fields, the signals for both orientations approach about half of the maximum signal. This behavior appears to be similar to some previous observations reported in nickel ferrite[5] and in YIG[42], which could be attributed to a mixture of the reduced magnon injection and the magnon scattering at domain walls. Due just to the efficiency of magnon injection and detection for a magnetization easy axis oriented 45° from the propagation direction, we would indeed expect

$R_{1\omega}/R_{1\max} = \sin^2(45°) = 1/2$. In contrast, when magnons propagate along the [110] direction, i.e. with Pt wires along the [1$\bar{1}$0] axis, $R_{1\omega}$ for $\phi = 90°$ decreases almost all the way to zero near the coercive fields, and $R_{1\omega}$ for $\phi = 0°$ stays constant near zero, indicating a full suppression of magnon transport (Fig. 3b). In this case, the magnetic moment orientations within the domains are perpendicular to the direction of spin polarization that is generated by the SHE and detected by the ISHE, therefore no spin transport signal is observed. Both of the geometries shown in Fig. 3a, b are therefore reasonably consistent with standard models of magnon propagation.

The results shown in Fig. 3c are unexpected, however. Remarkably, when magnons propagate along ([1$\bar{1}$0]), i.e. with Pt wires along the [110] axis, $R_{1\omega}$ reaches fully 95% of the saturated-state maximum for $\phi = 0°$. This angle corresponds to a magnetic sweep along the [110] direction, where a demagnetized domain-filled state exists at low field, with many 180° domain walls. Since the typical size of domains is smaller than the separation between the Pt wires (1 μm), this is a direct demonstration that the net spin polarization being transmitted does not reverse when the magnons pass through the 180° domain walls, because if that were the case the spin current would decrease drastically upon traversing even a single domain wall, on average. To verify the stability of magnon transport near zero field, we conducted measurements with $H_{ext}$ sweeping along different angles ($0° \leq \phi \leq 90°$). The results from these measurements (see Supplementary Note 3) show that regardless of the angle of the sweeping field, $R_{1\omega}$ always converges near zero field to the same value corresponding to nearly the maximum value for a saturated magnetic state.

We also measured the second harmonic signal for these different Pt wire orientations (lower panels of Fig. 3), which are expected to have a $\sin(\phi)$ dependence and are thus proportional to the average projection of the magnetization perpendicular to the Pt wires. This measurement should thus be analogous to a magnetization hysteresis curve for in-plane magnetic-field sweeps perpendicular to the Pt wires

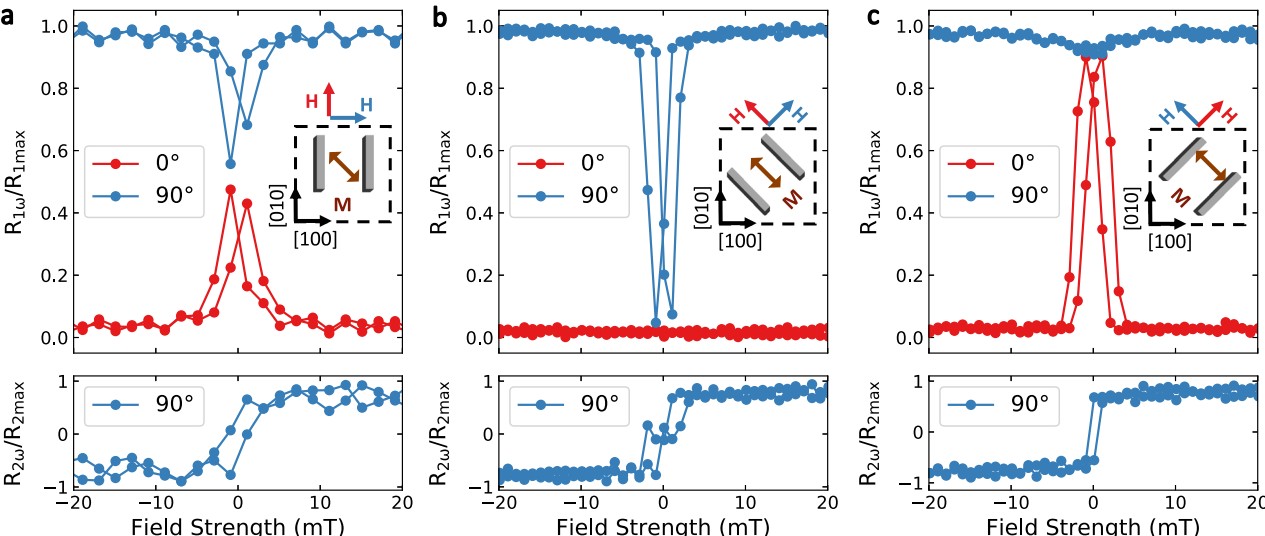

**Fig. 3 | Magnetic field dependence of nonlocal measurements.** Normalized nonlocal resistances for the Pt wires along **a** [010], **b** [1$\bar{1}$0], and **c** [110] axes for a wire separation of 1 μm. In each panel, the blue line corresponds to a magnetic-field scan perpendicular to the Pt wires and the red line corresponds to a field scan parallel to the Pt wires. For each of these cases, the upper (lower) panel shows the first (second) harmonic nonlocal resistance. We plot the second-harmonic curves only for $\phi = 90°$ because the second-harmonic signals for $\phi = 0°$ through the film to the detector where remain near zero in all cases.

(similar to Fig. 1a). This is what we observe. In particular, the second-harmonic data in Fig. 3b are consistent with the magnetization curve for the [110] sweep in Fig. 1a, with a plateau around zero field indicating a reorientation of the magnetization into the stripe domains along the [1$\bar{1}$0] easy axis. The second harmonic data in Fig. 3c also shows the same abrupt transition as the [1$\bar{1}$0] sweep in Fig. 1a. The fact that similar hysteresis loops are obtained for the SQUID measurements as well as the nonlocal second-harmonic spin transport measurements confirms the results from the NV center imaging, that domain formation is present throughout the MAFO film and is not just a local effect near the Pt electrodes.

For the case of Pt wires along [110] axis, where we have shown that spin angular momentum may diffuse through multiple domain walls without being suppressed, we investigated the dependence of the first-harmonic signal on the distance between the injector and detector Pt wires. Figure 4 shows the ratio of $R_{1\omega}$ around zero field to the maximum signal in the magnetically-saturated, spatially-uniform magnetic state, $R_{1\omega,max}$, as a function of this distance. We observe that the ratio stays close to 1, with no apparent distance dependence, at least up to the signal detection limit at around 4 μm. The first harmonic nonlocal resistance scans demonstrating this behavior for different wire separations are shown in the Supplementary Note 3. Based on the scanning NV center measurements, the average length of the domains along [1$\bar{1}$0] axis is smaller than 1 μm, so these measurements involve the traversal of more than 5 domain walls, on average.

## Discussion

As we have noted in the introduction of this paper, within conventional models for the interactions of a magnon with a magnetic domain wall it is very surprising that the presence of 180° domain walls produces a negligible effect on the nonlocal spin transport signals. Incoherently-excited magnons at room temperature should generally have short wavelengths compared to typical domain wall widths in MAFO. We would therefore expect the polarization of the spin carried by a magnon to reverse upon transmission through a 180° domain wall, and the net spin current carried by an incoherent population of magnons to decay quickly to zero in samples containing multiple domains[26]. (see Supplementary Note 7)

It is true that the room-temperature experiments we have performed differ in several aspects from simple independent-magnon

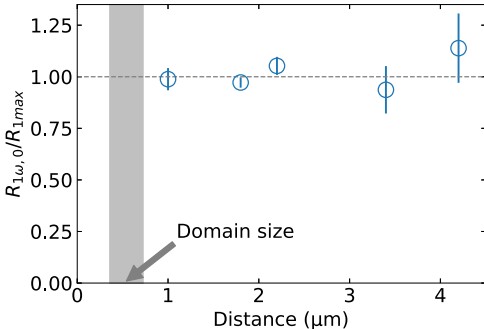

**Fig. 4 | Distance dependence of normalized nonlocal resistances in the multi-domain state.** The ratio of the first harmonic nonlocal resistance in the multi-domain state to the maximum nonlocal resistance in the spatially-uniform saturated magnetic state, plotted as a function of the distance between the Pt injector and detector wires. The Pt wires are oriented parallel to the [110] axis. The error bars indicate fitting uncertainties. Some of them are smaller than the size of the symbols. The gray bar indicates the range of typical domain sizes estimated from the magnetization map (Fig. 1b).

models. At room temperature, there is a substantial background of thermally-excited magnons, and magnon–magnon interactions cause the typical magnon mean free path to be quite short, <150 nm[43,44]. Therefore, assumptions of adiabatic passage through a domain wall without scattering are likely unrealistic. Unlike zero-temperature models, magnons in room-temperature magnets can also carry a net spin current with spin-moment direction parallel to the background magnetization direction as well as antiparallel, because magnon populations can be reduced below the room-temperature equilibrium value as well as enhanced above. Therefore, passage through a domain wall does not absolutely require that the direction of spin flow carried by diffusive magnons must be reversed. Nevertheless, given the strong exchange interaction between magnons and domain walls[28,30,31], we expect that no matter the value of the magnon mean free path a static 180° domain wall will exert substantial torque to rotate the spin direction of an incident diffusive spin current and should therefore suppress measurements of nonlocal spin currents. This is not what we observe.

We therefore suggest that a more exotic departure from the standard models of interactions between magnons and static domain walls is likely required in order to explain our experimental results. We have considered whether spin angular momentum might be carried by an overall motion of domain walls driven by interactions with the magnon currents, rather than entirely by magnons themselves. However, we do not observe any current-driven motion of domain walls in our NV center imaging (see Supplementary Note 4). We also are not aware of any process by which a net translation of domain walls could couple to the Pt detector wire to give an inverse spin Hall effect readout signal (see Supplementary Note 4). Nevertheless, our results might still be explained by taking into account the dynamics of domain walls, rather than assuming they are static. An assumption of static domain walls requires that angular momentum transferred from a spin current during adiabatic passage through a domain wall is immediately lost to the lattice. We speculate, instead, that even a domain wall that is pinned on average might not be completely static, but might undergo fluctuations that allow it to dynamically absorb angular momentum from an incident spin current and then reradiate most of that angular momentum back into the magnon population[45–49], rather than causing to be lost to the lattice. In this scenario, a domain wall would not be required to attenuate diffusive spin currents, but might absorb a spin current incident from one magnetic domain and re-emit it into a reversed domain without reversing the direction of spin polarization.

Another speculative scenario that may be worth considering is that angular momentum in nonlocal spin transport experiments might not be carried exclusively by magnons. Phonons can also transport angular momentum[50]. If they can do so with low damping, then phonons might be able to transport angular momentum through a magnetic domain wall without strong interaction. In this case, a magnon population excited near the source Pt electrode might transfer angular momentum to phonons, the phonons might ferry that angular momentum past a domain wall without reversing the angular momentum direction, and then the phonons might re-equilibrate with magnons on the far side before reaching the detector Pt electrode. To our knowledge, there is no evidence in the existing literature that angular momentum transport in magnetic insulators is carried exclusively by magnons and not phonons.

In summary, we have measured diffusive spin transport along different crystal axes in ultra-thin MAFO ferrimagnetic insulator films, for a multi-domain state within which a uniaxial magnetic anisotropy causes the sample to contain many 180° domain walls. We observed that the nonlocal spin transport signal in the multi-domain state can maintain at least 95% of the full value measured in the spatially uniform magnetically saturated state. The ratio of the signal in the multidomain state to the spatially-uniform state remains constant near 1 as a function of distance between the injector and detector electrodes, up to more than 5 times the average domain size. We argue that this result is incompatible with standard models for magnons interacting with static domain walls, in which the spin polarization carried by the magnon reverses upon passage through the domain wall. Instead, we suggest that to model these results one must consider the possibility of ingredients beyond magnons interacting with static domain walls – either that dynamic domain walls might absorb and re-emit magnons, or that phonons might carry angular momentum through 180° domain walls without inverting the angular momentum during passage.

## Methods

### Sample growth and fabrication
Epitaxial thin films of MAFO 6 and 10 nm thick were grown on $MgAl_2O_4$ (001) single crystal substrates by pulsed laser deposition using a stoichiometric $MgAl_{0.5}Fe_{1.5}O_4$ target. The details of the deposition conditions and the thin-film structural characterizations can be found in ref. 32. The Pt wires with dimensions 200 nm in width, 10 μm in length were defined using electron-beam lithography followed by a 30 W plasma cleaning and sputter-deposition of 10 nm Pt in an Ar atmosphere, and then lift-off. We patterned these Pt wires to be parallel and perpendicular to the magnetic easy axes of MAFO with various wire separations ranging from 0.4 to 4.2 μm. Electrical contacts to the Pt wires were made by the sputter-deposited Ti (5 nm)/Pt (100 nm) electrodes patterned using electron-beam lithography. The electrical resistance of each Pt wire is in the order of 5 kΩ.

### Nonlocal magnon transport measurements
The nonlocal spin transport measurements were performed at room temperature by using a lock-in amplifier with a low excitation frequency ($f = 5.939$ Hz) to minimize the inductive coupling between the wires. The current in the injector was applied with root mean squared (rms) amplitude ranging from 100 to 300 μA. With the presence of an applied in-plane magnetic field with desired strength and direction, the first and second harmonic voltage signals were detected simultaneously in the detector. This in-plane magnetic field were applied using a projected-field electromagnet which can be rotated 360°.

### Scanning NV center magnetometry measurements
The magnetic imaging is performed with a commercial scanning nitrogen-vacancy magnetometer (ProteusQ, Qnami AG) operating under ambient conditions. A commercial diamond tip hosting a single NV defect at its apex (Quantilever MX, Qnami AG) is integrated on a quartz tuning fork-based frequency modulation atomic force microscope (FM-AFM) and scanned above the MAFO sample surface with an NV flying distance of about ~60 nm, cf. Supplementary Fig. 9. The NV center orientation is characterized by the polar angle $\theta_{NV} = 57.1° \pm 2.5°$ and the azimuthal angle $\phi_{NV} = 270.3° \pm 0.9°$, respectively in the laboratory coordinates defined in Supplementary Fig. S9, and the NV depth is calibrated to be $d_{NV} = 59.7 \pm 1.8$ nm, following the method described in ref. 51.

## Data availability
The datasets supporting the conclusions are available in the article and the Supplementary Information, and also available from the corresponding authors on reasonable request.

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

## Acknowledgements

T.N. acknowledges support from the National Key R&D Program of China (Grant No. 2021YFA0716500), the National Natural Science Foundation of China (Grant No. 52161135103), and Tsinghua University Initiative Scientific Research Program. D.C.R. acknowledges support from the Cornell Center for Materials Research with funding from the NSF MRSEC program (Grant No. DMR-1719875). Y.S. acknowledges support from the Vannevar Bush Faculty Fellowship of the Department of Defense (contract No. N00014-15-1-0045). L.J.R. acknowledges support from the Air Force Office of Scientific Research (Grant No. FA 9550-20-1-0293) and an NSF Graduate Research Fellowship. J.H. acknowledges support from NSF grant CBET-2006028. This work was performed in part at the Cornell NanoScale Facility, a member of the National Nanotechnology Coordinated Infrastructure, which is supported by the NSF (Grant No. NNCI-2025233). The micromagneitc simulations were performed using Bridges at the Pittsburgh Supercomputing Center through allocation TG-DMR180076, which is part of the Extreme Science and Engineering Discovery Environment (XSEDE) and supported by NSF grant ACI-1548562. The authors thank Satoru Emori, Pu Yu, Dingfu Shao, and Evgeny Tsymbal for helpful discussions, Isaiah Gray and Gregory Fuchs for performing spin Seebeck imaging measurements, and CIQTEK for technical support for the NV center measurements (Diamond III QDAFM) with in-situ applied currents.

## Author contributions

T.N., R.L. and D.C.R. conceived the research. T.N., D.C.R., Y.S., J.H. and P.M. supervised the experiments. L.J.R., P.L., D.Y. and Y.S. performed the sample growth. R.L. and T.N. performed the device fabrication. R.L. and T.N. performed non-local magnon transport measurements and analysis. H.Z., Y.Z, D.A.B., A.E.E.D. and P.M. performed the scanning NV measurements and analysis. Y.C. and Y.Z. performed magnetic characterizations. M.D. and J.H. performed the micromagnetic simulations. B.L. performed the theoretical calculations. T.N., R.L. and D.C.R. wrote the manuscript. All authors discussed the results and commented on the manuscript. T.N. and D.C.R. directed the research.

## Competing interests

The authors declare no competing interests.
