## [Peer Review File · Nature Communications]

Reviewers' Comments:

Reviewer #1:

Remarks to the Author:

Li et al. presented their work on 'A puzzling insensitivity of magnon spin diffusion to the presence of 180° domain walls in a ferrimagnetic insulator' in manuscript number NCOMMS-22-15222. Some nice experimental results, including NV center magnetometry images and non-local transport measurements, are presented together with speculated cause. As the title suggested, the explanation is not very convincing based on the presented experimental results. A major revision will be needed before this manuscript can be considered to be published in Nature Communications.

Very unfortunately, the magnetic hysteresis loops and domain images were performed on 10 nm thick MAFO films but the non-local transport measurements on 6 nm ones. The strain and defect density can be very different for these two thicknesses, unless the authors can convince the readers. Magnetic proximity effect on Pt was not mentioned in the manuscript. Discussions on whether the magnetic proximity effect plays a role is necessary. In this sense, a control experiment with the Pt spin injector replaced by Al or Cu, the contact leads material, should be readily achievable.

For the non-local measurements, there are five data points in Fig. 4. However, only the data for the shortest separation between Pt wires, 1 micron apart, are presented in Figs. 2, 3, and S4. All data available should be presented in the supporting Information. Is it feasible to extract damping coefficient from the quantitative analysis of the distance dependence? This damping coefficient can be used in the simulation instead of the 0.0015 from the literature.

For the main topic, 'insensitivity of magnon through domain walls', more in depth discussions are expected. In addition to Ref. 20 already cited, which showed analytically a spin wave transmits through a domain wall without reflection and asserted the difference in spin angular momentum should be transferred to the domain wall, there are several relevant papers showing that the transmission of the spin waves through domain walls depends on the amplitudes and frequencies of the spin waves and the magnetization structures of the domain walls, as listed in the following (should be considered to be put in the reference).

1. Buijnsters, F. J., Ferreiros, Y., Fasolino, A. & Katsnelson, M. I. Chirality-dependent transmission of spin waves through domain walls. *Phys. Rev. Lett.* 116, 147204 (2016).
2. Chang, L.-J. et al. Ferromagnetic domain walls as spin wave filters and the interplay between domain walls and spin waves. *Sci. Reports* 8, 3910 (2018).
3. Hämäläinen, S.J., Madami, M., Qin, H. et al. Control of spin-wave transmission by a programmable domain wall. *Nat Commun* 9, 4853 (2018). <https://doi.org/10.1038/s41467-018-07372-x>

The authors of this manuscript should look into the details of their simulation results and discuss the domain structures. If the authors can show convincing data that magnons transmit between the source and detector with or without domain walls and believe there are no (static) current-driven domain wall motion as shown in Fig. S5, the dynamic domain wall argument together with domain structure analysis might be plausible.

Reviewer #2:

Remarks to the Author:

This manuscript reports an experimental study of magnon transport in ferrimagnetic thin films. The focus is on the ferrimagnetic insulator MgAlFeO (MAFO), which is grown epitaxially and possesses a multidomain state at zero field. By using spin Hall effects, the authors investigate magnon transport between two platinum electrodes deposited on the MAFO. A key finding is that little difference is found between magnon transport in the uniform state and in the multidomain state, where a reduction in magnon current is naively expected in the latter as the presence of domain walls should result in increased scattering. Some possible explanations for this observation are given in the concluding remarks of the manuscript.

While the results do appear to be striking, one has the impression that the work is more of a preliminary nature, rather than one backed by supplementary experiments, analyses, and supporting theoretical work. As such, I believe it is difficult to conclude at this stage whether the

observation really constitutes a departure from standard theory as the authors assert, or whether a more banal explication is yet to be found.

There are a number of points I didn't find very clear:

(i) In the multidomain state, I presume that the magnetic configuration is equally multidomain beneath the source and the detector (Pt wires). As such, how well defined is the magnon current polarity in this case, as presumably one would require some form of statistical averaging both at the source and detector regions?

(ii) Did the authors verify the multidomain state before and after (and possibly also during) the application of the currents (spin Hall effect)? If the magnon currents do interact with the domain structure, one might expect to see some shifts in the domain states, or changes in domain sizes, which might influence the overall transport.

(iii) Given the multi-sublattice nature of ferrimagnets, to what extent are magnon transport theories based on ferromagnets applicable here? If we take the other extreme case, i.e., that for antiferromagnets, then would magnon transport theories interpolate smoothly between the two cases?

Concerning the basic magnon-domain wall theory, a number of pioneering works have been left out of the reference list, where the special solutions of the spin wave eigenmodes within 180 degree Bloch domain walls, which act as reflectionless potentials, have been discussed, along with their influence on wall motion:

J. M. Winter, Phys. Rev. 124, 452 (1961)

A. V. Mikhailov and A. I. Yaremchuk, JETP Lett. 39, 357 (1984)

D. Bouzidi and H. Suhl, Phys. Rev. Lett. 65, 2587 (1990)

H.-B. Braun, Phys. Rev. B 50, 16485 (1994)

Overall, I believe the manuscript contains some potentially new and exciting results, but the lack of theoretical analysis and complementary, supporting measurements means that the work is probably too preliminary to be considered for publication in Nature Communications at this stage.

Response to Reviewers

We appreciate the Reviewers' interest in this work and their comments to help improve the manuscript. Both reviewers acknowledged the novelty of our experimental results while posing some questions about different aspects. We have addressed all of their questions.

We have carefully considered all of the Reviewers' comments and modified the manuscript accordingly. The Reviewers' comments are reproduced below, and our responses are in the blue text. We also include a revised manuscript that contains editing markups so that our changes are easily identified.

REVIEWER 1 COMMENTS:

Li et al. presented their work on 'A puzzling insensitivity of magnon spin diffusion to the presence of 180° domain walls in a ferrimagnetic insulator' in manuscript number NCOMMS-22-15222. Some nice experimental results, including NV center magnetometry images and non-local transport measurements, are presented together with speculated cause. As the title suggested, the explanation is not very convincing based on the presented experimental results. A major revision will be needed before this manuscript can be considered to be published in Nature Communications.

The main conclusions of the paper are (1) the experimental result that magnon spin transport in MAFO is largely insensitive to the presence of 180° domain walls and (2) that this result disagrees with previous theories about what should happen to the spin angular momentum carried by a magnon when it encounters such a domain wall. We argue that the evidence and argument that we provide to justify these main conclusions are convincing. However, the Reviewer does not appear to provide any evaluation at all about these main conclusions of the paper. Instead, the Reviewer merely criticizes as "not very convincing" remarks that we already clearly label as speculative. While we think these speculative considerations are important for sparking discussion in the community about what assumptions of the existing theory might be incorrect, they are not the main conclusions of the paper. We therefore suggest that the Reviewer is applying incorrect standards in the review, and we would ask that the Reviewer to instead address the main conclusions of the paper.

- Very unfortunately, the magnetic hysteresis loops and domain images were performed on 10 nm thick MAFO films but the non-local transport measurements on 6 nm ones. The strain and defect density can be very different for these two thicknesses, unless the authors can convince the readers.

In our original submission, the NV center measurements were performed on the 6 nm thin film, while magnetic hysteresis loops were measured on the 10 nm film due to the stronger magnetic signal. We have now also measured magnetic hysteresis loops for the 6 nm film as shown below, using vibrating-sample magnetometry (VSM). The hysteresis loops of the 6 nm sample are very similar to the 10 nm sample. We now include Figure R1 in our main text (as Fig. 1a), and we have moved the data for the 10 nm film to the supporting information. We have also included in the supporting information new nonlocal magnon transport measurements for the 10 nm film, which show results very similar to those for the 6 nm film.

FIG. R1. Magnetic hysteresis loops of the 6 nm film along the $[110]$ (blue) and $[1\bar{1}0]$ (red) directions, measured with VSM.

- Magnetic proximity effect on Pt was not mentioned in the manuscript. Discussions on whether the magnetic proximity effect plays a role is necessary. In this sense, a control experiment with the Pt spin injector replaced by Al or Cu, the contact leads material, should be readily achievable.

The magnetic proximity effect is not relevant to our conclusions, as the Pt is not present in the region through which the magnons are traveling. In addition, since the first-harmonic nonlocal spin signal that we analyze is generated by spin Hall effect, changing the spin injector material to Al or Cu (with a negligible spin Hall effect) would simply eliminate our signal rather than providing useful information.

- For the non-local measurements, there are five data points in Fig. 4. However, only the data for the shortest separation between Pt wires, 1 micron apart, are presented in Figs. 2, 3, and S4. All data available should be presented in the supporting Information.

The magnetic field dependences of the nonlocal resistance data for the other spacings are now added in the supporting information and also included below:

FIG. R2. First-harmonic nonlocal resistances for for same 6 nm MAFO samples discussed in the main text with a wire separation of (a) 1.8 μm , (b) 2.2 μm , (c) 3.4 μm and (d) 4.2 μm .

- Is it feasible to extract damping coefficient from the quantitative analysis of the distance dependence? This damping coefficient can be used in the simulation instead of the 0.0015 from the literature.

It is not straightforward to extract the damping coefficient from the magnon spin diffusion measurements, since the spin diffusion length also depends on other factors *e.g.* magnetic anisotropy, as reported in our recent paper (Nano Lett. 2022, 22, 3, 1167–1173). Instead, to measure the damping coefficient of our sample we have now performed ferromagnetic resonance (FMR) measurements. Figure R3 shows the FMR linewidth as a function of frequency, from which the damping coefficient is determined to be 0.004. We have redone the micromagnetic simulation with the experimentally-measured damping coefficient and we find that the damping coefficient has little influence on the results, because the simulation is mainly to demonstrate the direction of magnetization. We have included new simulation results below and in the supplementary information.

FIG. R3. Frequency dependence of FMR linewidth for the 6 nm MAFO film with the field applied along the [110] direction.

FIG. R4. Domain pattern in the demagnetized state obtained via micromagnetic simulations with damping coefficient = 0.004.

- For the main topic, ‘insensitivity of magnon through domain walls’, more in depth discussions are expected. In addition to Ref. 20 already cited, which showed analytically a spin wave transmits through a domain wall without reflection and asserted the difference in spin angular momentum should be transferred to the domain wall, there are several relevant papers showing that the transmission of the spin waves through domain walls depends on the amplitudes and frequencies of the spin waves and the magnetization structures of the domain walls, as listed in the following (should be considered to be put in the reference).
 - Buijnsters, F. J., Ferreiros, Y., Fasolino, A. & Katsnelson, M. I. Chirality-dependent transmission of spin waves through domain walls. *Phys. Rev. Lett.* 116, 147204 (2016).
 - Chang, L.-J. et al. Ferromagnetic domain walls as spin wave filters and the interplay between domain walls and spin waves. *Sci. Reports* 8, 3910 (2018).

3. Hämäläinen, S.J., Madami, M., Qin, H. et al. Control of spin-wave transmission by a programmable domain wall. *Nat Commun* 9, 4853 (2018). <https://doi.org/10.1038/s41467-018-07372-x>

Thanks for providing these helpful references; we have added them to the paper. These papers listed by the Referee generally focus on how the transmission of magnons depends on the domain wall configuration, e.g, Neel wall or Bloch wall, 90° or 180° wall. Our analysis pertains to 180° domain walls, since the uniaxial anisotropy of our samples enforces this configuration. Previous studies have shown that a spin wave with a wavelength shorter than the width of the domain wall is expected to transmit through a 180° domain wall without reflection while reversing its spin orientation (please see PRL 107, 177207 (2011), PRL 112, 147204 (2014), PRB 100, 174403 (2019)). Therefore, our statements about angular momentum transfer from the magnon to the domain wall due to spin flip are well-founded. Furthermore, in the revised supporting information, we also provide a more detailed analytical discussion about spin waves in a ferrimagnet interacting with a static 180° Neel domain wall and show how this can be reflectionless.

As noted by the Referee, the degree to which spin wave transmission is reflectionless depends on the spin-wave wavelength, with partial reflection becoming significant when the wavelength is comparable to or larger than the domain wall width. We expect that our signals are dominated by wavelengths shorter than this crossover. Given an exchange interaction of $A = 2.2 \times 10^{-12}$ J/m and a uniaxial anisotropy of $K_{uni} = 4 \times 10^3$ J/m³, our samples should have a wall width $\sqrt{A/K_{uni}} \approx 24$ nm, so that only wavevectors with magnitude less than 0.28 nm^{-1} will correspond to wavelengths longer than the domain wall width. We estimate that this represents only about 3% of the modes that are active at room temperature. (See the supplementary material.)

- The authors of this manuscript should look into the details of their simulation results and discuss the domain structures. If the authors can show convincing data that magnons transmit between the source and detector with or without domain walls and believe there are no (static) current-driven domain wall motion as shown in Fig. S5, the dynamic domain wall argument together with domain structure analysis might be plausible.

In the revised supplemental material, the analytical form of domain wall structure is given based on experimental and simulation results. We have also performed new NV center imaging experiments which demonstrate that there is no significant magnon-driven long-range translation of domain walls in our samples. (See the discussion below for comment ii from Reviewer 2 and the associated images in Fig. R5.)

REVIEWER 2 COMMENTS:

This manuscript reports an experimental study of magnon transport in ferrimagnetic thin films. The focus is on the ferrimagnetic insulator MgAlFeO (MAFO), which is grown epitaxially and possesses a multidomain state at zero field. By using spin Hall effects, the authors investigate magnon transport between two platinum electrodes deposited on the MAFO. A key finding is that little difference is found between magnon transport in the uniform state and in the multidomain state, where a reduction in

magnon current is naively expected in the latter as the presence of domain walls should result in increased scattering. Some possible explanations for this observation are given in the concluding remarks of the manuscript. While the results do appear to be striking, one has the impression that the work is more of a preliminary nature, rather than one backed by supplementary experiments, analyses, and supporting theoretical work. As such, I believe it is difficult to conclude at this stage whether the observation really constitutes a departure from standard theory as the authors assert, or whether a more banal explication is yet to be found.

There are a number of points I didn't find very clear:

(i) In the multidomain state, I presume that the magnetic configuration is equally multidomain beneath the source and the detector (Pt wires). As such, how well defined is the magnon current polarity in this case, as presumably one would require some form of statistical averaging both at the source and detector regions?

What is important is the sign of the flow of spin angular momentum, and this will be the same in both types of domain, with magnetization either aligned or anti-aligned with the spin moment from the spin Hall effect. For a ferromagnet, if a domain has magnetization anti-aligned with the spin moment from the spin Hall effect, the result is an increased number of magnons compared to the thermal background. For the opposite domain, the result is a decrease in the number of magnons flowing compared to the thermal background. However, since the spin angular momentum carried by each magnon mode is opposite for the two domains, the spin current excited by the spin Hall effect ends up having the same sign in both domains. In the case of a ferrimagnet (as opposed to a ferromagnet), magnon populations with spins both parallel and antiparallel to the background magnetization may contribute within each domain, but the overall conclusion is unchanged. This can be seen from the experimental data in Fig. 2c in which a saturating magnetic-field is rotated in-plane: the first-harmonic signal generated by the spin Hall effect is the same upon rotating the magnetization by 180° .

(ii) Did the authors verify the multidomain state before and after (and possibly also during) the application of the currents (spin Hall effect)? If the magnon currents do interact with the domain structure, one might expect to see some shifts in the domain states, or changes in domain sizes, which might influence the overall transport.

We carefully studied the possible current-induced magnetic domain wall motions by using NV center magnetometry with in-situ current pulses. Figure R5(a), (b) and (c) show the magnetic field map measured by the NV center with no current and after applying several 0.2 mA or -0.2 mA current pulses, respectively. We also used the same current amplitude in our magnon transport measurements (Fig. S5). The NV center scanned area is about $0.8 \mu\text{m}$ away from the center of Pt wire with its direction along [110] direction. This is in the same configuration as that of Fig. 3c, where the spin current can transport between the two Pt bars in the multidomain state. However, we do not observe any significant change in the domain pattern between scans.

FIG. R5. (a) Schematic diagram of the setup to apply current pulses for the 6 nm MAFO film, and comparison of the NV-center images due to the magnetic domain pattern (b) before and (c) (d) after applying 0.3 s long current pulses.

(iii) Given the multi-sublattice nature of ferrimagnets, to what extent are magnon transport theories based on ferromagnets applicable here? If we take the other extreme case, i.e., that for antiferromagnets, then would magnon transport theories interpolate smoothly between the two cases?

In the revised supplemental material, we provide more details on magnon theory in a domain wall and relevant discussion of possible transport mechanism. Based on our analysis and the papers recommended by the referee below, if a magnon transmits without reflection through a 180° domain wall and magnon-magnon interactions can be neglected then the magnon will flip its polarization upon passing through the domain wall. This holds regardless of whether one considers a ferromagnet, ferrimagnet, or antiferromagnet (please see PRL 107, 177207 (2011), PRL 112, 147204 (2014), PRB 100, 174403 (2019)). The difference between a ferromagnet and ferrimagnet (or antiferromagnet) is that magnons in the former case only have one possible polarization direction in each domain, while the latter case there are two branches of magnons with opposite polarizations. Nevertheless, for both ferromagnets and ferrimagnets, a magnon transmitting through a static domain wall without magnon-magnon interactions is expected to reverse its spin polarization following the texture of the domain wall. Furthermore, any direct interaction or scattering between the two branches of magnons in a ferrimagnet would degrade the flow of angular momentum, rather than allowing efficient transmission of the angular momentum through a domain wall. What is required to explain our experiment is a process that allows passage of excitations through the domain wall while conserving the angular momentum in the excitations.

- Concerning the basic magnon-domain wall theory, a number of pioneering works have been left out of the reference list, where the special solutions of the spin wave eigenmodes within 180 degree Bloch domain walls, which act as reflectionless potentials, have been discussed, along with their influence on wall motion:
- J. M. Winter, Phys. Rev. 124, 452 (1961)
- A. V. Mikhailov and A. I. Yaremchuk, JETP Lett. 39, 357 (1984)
- D. Bouzidi and H. Suhl, Phys. Rev. Lett. 65, 2587 (1990)
- H.-B. Braun, Phys. Rev. B 50, 16485 (1994)

Thanks for providing these references, which we have included in the revised submission. We do note that these papers all consider the effects of Bloch domains. For our samples, the domain walls should be of the Néel type, but this does not change the physics. In our revised supplemental materials, we have included a calculation of the eigenmodes for magnons in a 180° Néel domain wall, which shows the same reflectionless characteristic.

- Overall, I believe the manuscript contains some potentially new and exciting results, but the lack of theoretical analysis and complementary, supporting measurements means that the work is probably too preliminary to be considered for publication in Nature Communications at this stage.

Most experimental papers confirm what is already expected. It is more important when a paper shows that the conventional understanding is incorrect or incomplete. Our paper is in that category. We also take issue with the description of our work as “preliminary.” We establish an unambiguous experimental result that when our samples contain many domain walls (based on magnetometry and NV microscopy) that the magnon spin transport signals are not attenuated as predicted by established theory. This shows the need for significant corrections to understanding in this field. This conclusion of our paper is not in any way preliminary in the meaning of being tentative or requiring further data to establish.

The main criticism of Reviewer 2 appears to be that perhaps there is a “banal explication” for what we measure, but this criticism is made without offering even a suggestion about what that explication might be. Our result is exceedingly straightforward – the presence of 180° domain walls has almost no effect on nonlocal spin transport, in stark contrast with existing expectations for what should happen to the spin angular momentum when magnons encounter a domain wall. We urge the Reviewer not to casually dismiss this result without considering it carefully and trying to propose an explanation. We argue that our results necessitate a reconsideration of fundamental aspects of nonlocal spin transport with a degree of importance that fully justifies publication in Nature Communications.

Reviewer #1 (Remarks to the Author):

Li et al. revised and resubmitted their work on 'A puzzling insensitivity of magnon spin diffusion to the presence of 180° domain walls in a ferrimagnetic insulator'. By including Supplementary Note 7, they presented more completed description of related fields. The authors showed strong bias against the one-dimensional model showing 'Reflectionless transmission of spin waves expected through static domain walls', though it is in favor of their experimental results. One possible reason I think is that it is hard to argue what exactly is measured across the 10-micron long Pt spin detector.

Since the experimental data are solid and are important for sparking discussion in the community, I would recommend the manuscript to be accepted after minor revision. The authors should present existing theories in equal footing, then express their opinions in the discussion. Some more quantitative discussion on the energy scales in concern will help.

1. The words 'A puzzling' should be removed from the title;
2. The main points in Supplementary Note 7 need to be stated in the main text;
3. Rewrite the manuscript with no bias on different models.
4. More information in Supplementary Note 6 should be provided.

Response to Reviewers

We appreciate the Reviewer's comments to help improve the manuscript. We have carefully considered all the comments and modified the manuscript accordingly. The Reviewer's comments are reproduced below, and our responses are in the blue text. Given that the paper has undergone two rounds of review over 11 months, we hope that you will now be able to accept it for publication without additional review.

Li et al. revised and resubmitted their work on 'A puzzling insensitivity of magnon spin diffusion to the presence of 180° domain walls in a ferrimagnetic insulator'. By including Supplementary Note 7, they presented more completed description of related fields. The authors showed strong bias against the one-dimensional model showing 'Reflectionless transmission of spin waves expected through static domain walls', though it is in favor of their experimental results. One possible reason I think is that it is hard to argue what exactly is measured across the 10-micron long Pt spin detector.

Since the experimental data are solid and are important for sparking discussion in the community, I would recommend the manuscript to be accepted after minor revision.

We appreciate Reviewer #1's positive recommendation for our paper to be accepted, but we are confused by the statement that the conventional picture of transmission of spin waves through static domain walls is "in favor of their experimental results". The message of our paper is that our measurements are in conflict with the conventional model. In the conventional picture, the angular momentum carried by spin waves rotates by 180° as they travel within the exchange field of a static 180° domain wall, so the signal from the diffusive spin current would not remain unchanged, with the same sign and magnitude, after traveling a distance much larger than a typical domain size (as we observe).

The authors should present existing theories in equal footing, then express their opinions in the discussion.

See point #3 below.

Some more quantitative discussion on the energy scales in concern will help.

The magnon energies in MAFO, like other oxide magnets, range from close to zero to about 100 meV, so at room temperature there are thermal magnons excited over a large fraction of the Brillouin zone. The paper already states, "At room temperature, there is a substantial background of thermally-excited magnons." We have also already provided in Supplementary Note 7 a quantitative discussion of the magnon wavevectors that can be excited at room temperature. The phonons have a similar range of energies and active wavevectors.

1. The words ‘A puzzling’ should be removed from the title;

The Reviewer does not give any reason why our wording is incorrect, so this seems to be merely a stylistic preference. In our view, the words “A puzzling” convey the important information that our measurements are in conflict with the commonly-used picture of spin-wave transport in magnetic materials, which is the primary reason why our paper is of interest. It is our strong preference to retain “A puzzling” within the title to encourage readers to study our paper and to promote discussion about how this commonly-used picture of spin-wave transport must be modified or augmented to explain our results.

2. The main points in Supplementary Note 7 need to be stated in the main text

Supplementary Note 7 provides a detailed discussion of the reflectionless transmission theory for spin waves. The purpose of this Note is to show, in the traditional picture of adiabatic passage through a 180° domain wall, that magnons always switch polarization direction, which should suppress the spin current signal. We have already summarized this point in the main text: “We would therefore expect the polarization of the spin carried by a magnon to reverse upon transmission through a 180° domain wall, and the net spin current carried by an incoherent population of magnons to decay quickly to zero in samples containing multiple domains [24].” We have now also added a parenthetical remark at the end of this sentence “(see Supplementary Note 7)” in order to direct readers who are interested in more details.

3. Rewrite the manuscript with no bias on different models.

We are confused here what are the “different models” for which the Reviewer wishes that there should be no bias. As we explained above, the main message of our paper is that our measurements are in conflict with the conventional picture of spin wave transmission through static domain walls. So this cannot be one of the models we present with no bias. Near the end of the paper, we speculate about two possible extensions of the conventional model that might explain our results – that the domain walls might not be static or that angular momentum might be carried by excitations other than spin waves. We believe that this part of our discussion already complies with the Reviewer’s desire for “no bias” in that we do not present one possibility as more probable than the other.

4. More information in Supplementary Note 6 should be provided.

We have now added the raw data of ferromagnetic resonance spectra taken at different frequencies and the description on how the Gilbert damping coefficient was estimated, in Supplementary Note 6.